# Unexplained Left Ventricular Hypertrophy with Symptomatic High-Grade Atrioventricular Block in Elderly Patients: A Case Report

**DOI:** 10.3390/jcm11123522

**Published:** 2022-06-19

**Authors:** Tzu-Ping Yu, Ju-Yi Chen

**Affiliations:** Department of Internal Medicine, National Cheng Kung University Hospital, College of Medicine, National Cheng Kung University, Tainan 704, Taiwan; g860920@gmail.com

**Keywords:** unexplained hypertrophy, elderly, Fabry disease, high-grade atrioventricular block

## Abstract

Left ventricular hypertrophy (LVH) is common among older adults. Amidst all causes, Fabry disease (FD) should be considered when LVH occurs with family history, specific clinical manifestations, or cardiac alert signs. Here, we report a case of a 76-year-old male who presented late onset concentric LVH with symptomatic high-grade atrioventricular (AV) block. After dual-chamber pacemaker implantation, interrogation revealed frequent right ventricular (RV) pacing with a wide QRS duration. The patient developed heart failure symptoms with rapid deterioration of LV systolic function. Pacing-induced cardiomyopathy (PICM) was suspected, and the pacemaker was upgraded to biventricular pacing. Further FD surveys were performed, including biochemical examinations, cardiac biopsies, and genetic sequencing, and the patient was ultimately diagnosed with a cardiac variant of FD. Particularly, we strongly suggest that physiologic pacing should be initially considered for patients with FD who have symptomatic high-grade AV block, rather than traditional RV pacing to prevent PICM.

## 1. Introduction

Left ventricular hypertrophy (LVH) is a common, but ominous, discovery in older adults that is associated with an increased risk of cardiovascular morbidity and mortality. Although hypertension, valvulopathies, and obesity elucidate most causes of LVH, it is crucial to consider other rare causes if unexplained LVH persists. Less common causes of LVH include numerous myocardial disorders, such as hypertrophic cardiomyopathy (HCM), infiltrative diseases, metabolic disorders, mitochondrial diseases, and some syndromic conditions [1,2,3]. Amidst all possible causes, Fabry disease (FD) should be considered when LVH is accompanied by a family history of FD, specific clinical manifestations of FD, or certain cardiac alert signs [4,5,6]. Clinicians have discussed specific cardiac alert signs of FD intensely, which may possibly reveal more clues and increase cardiologists’ vigilance in investigating FD. FD is an inherited lysosomal storage disorder that results in multisystem diseases. The reported prevalence of FD varies widely, ranging from 1:17,000 to 1:117,000 [4,6]. Cardiovascular complications are the leading cause of impaired quality of life and death in all FD patients [4,5,6]. Early detection of FD before irreversible organ damage occurs, as well as the prompt initiation of effective treatment, are considered extremely important. Nonetheless, the diagnosis of FD is burdensome because the full picture of the disease is not yet recognized, and caution is lacking regarding the FD-related symptoms or signs. The final diagnosis of FD is usually made years after the onset of primary symptoms or signs, and many cases have been greatly underdiagnosed.

Herein we report a case of a 76-year-old male who was diagnosed as FD with symptomatic high-grade atrioventricular (AV) block. Furthermore, we have proposed an algorithm by which to evaluate patients with unexplained LVH, in order to support the diagnostic and therapeutic management of FD with high-grade AV block.

## 2. Case Presentation

In 2008, a 63-year-old male with no known underlying diseases was referred due to recurrent palpitation. A 12-lead electrocardiogram (ECG) showed sinus rhythm with ventricular pre-excitation (Figure 1A). Seeking a link between palpitations and abnormal rhythm, the 24-h Holter monitor described no tachyarrhythmia. Transthoracic echocardiogram (TTE) showed concentric left ventricular hypertrophy (LVH) with adequate systolic function (Figure 1D,E).

From 2008 to 2015, the patient was repeatedly referred due to recurring unexplained palpitations and occasional retrosternal oppression. Holter monitor and TTE continued to have similar findings from his primary reports in 2008 (Table 1). In 2016, he visited the emergency department once, owing to episodic dizziness with near syncope. A 12-lead ECG showed bradycardia (37 bpm) and 3:1 atrioventricular (AV) block (Figure 1B). Emergent temporary transvenous pacing was performed, and then he was admitted for permanent pacemaker implantation (Medtronic, DDDR mode). Following discharge, he was followed regularly by his private cardiologist. Since 2017, his pacemaker interrogation reports have recorded right ventricular (RV) pacing, dependent as predominant rhythm without prominent symptoms. From 2019 to 2021, he began to feel dyspnea on exertion and increasingly aggravated, which met the criteria for New York Heart Association (NYHA) functional class III. TTE revealed impaired left ventricular (LV) systolic function (LV ejection fraction [EF] 20%). Under the impression of heart failure with reduced ejection fraction, standard treatments were prescribed and titrated to the optimized dosage (drugs: carvedilol, furosemide, ramipril, and spironolactone). However, he was still symptomatic, and rapid deterioration of LV systolic function was noted by TTE, with widening of pacing QRS on ECG (Table 1). Hence, pacing-induced cardiomyopathy (PICM) was assumed, so that the upgrading of his permanent pacemaker to cardiac resynchronization therapy (CRT: biventricular pacemaker) was executed. After upgradation, both TTE and ECG (Table 1) displayed markedly improved heart function, and his symptoms subsided, thus supporting the diagnosis of PICM.

On the other hand, because of his constantly unexplained concentric LVH with intensified symptoms, extra investigations were launched. After excluding the common secondary causes of LVH, such as valvular heart diseases, systemic hypertension, and obesity, other rare diseases, including familial HCM, infiltrative diseases, and metabolic storage disorders, were under-differentiated. According to his clinical manifestations, ECG and TTE interpretations, FD was initially suspected. Accordingly, we began comprehensive FD survey (Table 2). However, the patient denied family history of FD, and no extracardiac presentations had been revealed. In accordance with the entire results of his FD survey, this patient was consequently diagnosed with cardiac variants of FD, and further received enzyme replacement therapy (agalsidase beta 1 mg/kg every other week).

## 3. Discussion

### 3.1. Differential Diagnosis of Elderly LVH

In older adult patients, LVH can develop from primary cardiomyopathy or secondary to extrinsic stimuli (pressure or volume overload). Extrinsic stimuli are the most common causes, including systemic hypertension, valvular diseases (e.g., aortic stenosis), and obesity, which all need to be carefully excluded first. For unexplained LVH, rare myocardial disorders, including HCM, infiltrative diseases (e.g., amyloidosis and sarcoidosis), metabolic disorders (e.g., FD, Pompe disease, and *PRKAG2* syndrome), and mitochondrial diseases, should be thoroughly investigated. In the present case, the patient presented with late onset LVH, with symptomatic high-grade AV block and without extracardiac comorbidities, suggesting that HCM, cardiac amyloidosis, and FD were the most probable causes. Other rare diseases (e.g., sarcoidosis and mitochondrial diseases) usually develop at earlier ages and tend to present broad extracardiac features [3]. Pompe disease can sometimes be late onset but is often characterized by progressive skeletal muscle weakness or loss of respiratory function [8]. PRKAG2 syndrome usually displays LVH with conduction abnormalities (e.g., ventricular pre-excitation), yet onset is mostly in childhood. As mentioned above, we prioritized HCM, cardiac amyloidosis, and FD when considering the patient’s differential diagnosis.

***Hypertrophic cardiomyopathy (HCM).*** HCM is the most common inherited cardiomyopathy and relates to imperative cause of sudden cardiac death (SCD), especially the obstructive type. As such, it should be placed at high hierarchy in the scheme of elderly LVH. HCM typically manifests as the early onset of asymmetric septal hypertrophy but could be late onset and highly variable (e.g., apical, concentric, and right ventricular hypertrophy). Favorable determinants of HCM are family history of HCM or SCD, as well as dynamic LV outflow tract obstruction or systolic anterior motion of mitral valve presenting on echocardiogram [1]. Advanced diagnostic methods include cardiac biopsy (myofibrillar disarray and fibrosis) and genetic sequencing (genetic mutations in sarcomere or sarcomere-associated proteins). Diagnosis of HCM is challenging, given phenotypic heterogeneity, numerous unknown genetic mutations, and often a clinical diagnosis of exclusion [1].

***Cardiac amyloidosis.*** As for cardiac amyloidosis, the more likely cause of elderly LVH is wild-type transthyretin amyloidosis (ATTRwt), previously called age-related amyloidosis. ATTRwt gives rise to amyloidotic cardiomyopathy, resulting from the deposition of misfolded or misassembled transthyretins that are prone to form amyloid fibrils aggregates within the myocardium. The majority of patients typically present with diastolic heart failure or atrial fibrillation in old age. ECG findings, such as low voltage, pseudoinfarct pattern, or AV block, may possibly be seen. Echocardiogram has demonstrated a generalized concentric hypertrophy, biatrial dilation, and mainly diastolic dysfunction [9]. Establishing a diagnosis of ATTRwt is difficult, due to the lack of definitive biomarkers, and the clinical characteristics frequently mimic the diseases coexisting in advanced age, although cardiac scintigraphy with 99 m Tc-diphosphonates may play a useful role in the diagnosis of ATTRwt [10]. Final diagnosis is usually made by cardiac biopsy, which directly identifies the amyloid fibrils using Congo red staining and specifies the type by immunohistochemistry or mass spectrometry.

***Fabry disease (FD).*** FD is an X-linked recessive genetic disease, resulting in insufficient activity of lysosomal enzyme, α-galactosidase A (α-Gal A), which causes accumulation of glycosphingolipids, especially globotriaosylceramide (Gb3) and its deacylated derivative globotriaosylsphingosine (lyso-Gb3). These cumulative sediments within cells may eventually cause organ destruction (Figure 2) [5]. Typical FD clinical manifestations contain extracardiac involvements, comprising hypo- or anhidrosis, angiokeratomas, cornea verticillata, gastrointestinal manifestations, renal insufficiency, premature stroke, neuropathic pain, tinnitus or hearing impairment, and cardiac involvements (i.e., Fabry cardiomyopathy), including LVH, heart failure and conduction abnormalities. Classic FD symptoms normally present during early childhood but can be delayed in heterozygous cases. A late onset phenotype (i.e., cardiac variants) presents chiefly cardiac involvements, especially LVH, as a primary sign after the fourth decade of life [3,5,11,12]. The disease course of this phenotype is still largely unknown but tends to develop severe cardiac diseases in elderly individuals [11]. According to numerous studies, FD should be considered when unexplained LVH is found in combination with specific cardiac alert signs recorded by ECG, echocardiography, or cardiac magnetic resonance imaging (CMR) (Table 3) [3,4,5]. Both family history of FD and extracardiac presentations should be carefully scrutinized. Nevertheless, the lack of extracardiac manifestations and late onset phenotype may account for the absence of family history of FD. The earliest cardiac signs are subtle ECG changes, as well as LVH patterns with repolarization abnormalities or short PR interval. In advanced cases, sinus bradycardia, high voltage QRS, conduction disturbances, T wave inversion, or ST- segment deviation may be observed. FD patients are at relatively high risk of developing conduction diseases (e.g., any degree of AV block and arrhythmia). Consequently, regular 24-h Holter monitoring is recommended. Echocardiography is the most crucial instrument for initial diagnosis and monitoring of Fabry cardiomyopathy. The early stage is characterized by concentric ventricular hypertrophy without LV outflow tract (OT) obstruction, for which the LV systolic function is generally normal. Other particular findings may be noted, including prominent papillary muscles, early diastolic dysfunction, and right ventricular hypertrophy. As Fabry cardiomyopathy progresses, a regional wall (esp. basal inferolateral wall) hypo- or akinesis may develop. The non-invasive gold standard for detecting FD is CMR with gadolinium contrast agents. It provides accurate assessment of LV size, mass, and myocardial fibrosis involvement. Representative CMR features include late gadolinium enhancement (LGE) distinctively starting from the basal inferolateral wall and reduced T1 values in the affected area [4,5,13].

Relative to the present case, HCM, cardiac amyloidosis (ATTRwt), and FD (cardiac variants) may likely be causes, in line with the patient’s clinical presentations. Therefore, we differentiated between HCM, ATTRwt, and cardiac variants of FD, in terms of image interpretations (ECG and TTE). According to his initial ECG showing, as well as LVH with pre-excitation pattern and without low voltage QRS or pseudoinfarct pattern (Figure 1A), cardiac variants of FD was the most suspicious culprit. According to his echocardiography, which revealed generalized ventricular hypertrophy without LVOT obstruction (Figure 1D,E), this feature seemed to comply with cardiac amyloidosis or FD but was atypical for HCM. In brief, both cardiac variants of FD and ATTRwt were the likely suspects and held for further investigation. To distinguish these two diseases, biochemical blood tests for FD were first arranged due to currently available non-invasive examinations. Finally, cardiac biopsy and genetic sequencing were performed to confirm diagnosis. After thoroughly tracing the patient’s clinical history (Table 1) and clarifying the clinical findings, this patient clearly could have been diagnosed with a cardiac variant of FD since 2008.

### 3.2. Diagnosis of FD

Patients with suspicion of FD should undergo certain biochemical and genetic examinations (Table 3) [4,5,6]. In general, male patients with FD typically have reduced or absent α-Gal A activity and elevated lyso-Gb3 level, as measured in dried blood spot or blood leukocytes, which are adequate to confirm FD diagnosis. However, in females, X-chromosome inactivation results in widely variable clinical phenotypes and α-Gal A activity may be normal or slightly defective. Hence, the diagnosis of FD in females requires genetic sequencing. On the whole, all FD diagnoses should eventually be verified by genetic sequencing, which is helpful for establishing the relationship between the disease phenotype and genotype, and further permits cascade screening for high-risk family members [4,5,7]. Endomyocardial biopsy may be performed in patients with genetic variants of unknown significance, or under unusual phenotypic manifestations, which offer definitive evidence of FD by demonstrating lace-like appearance, vacuolization, and representative lysosomal inclusions (zebra bodies) on electron microscopy [4].

### 3.3. Management of FD

To date, the management of FD includes disease-specific therapy, as well as therapies to manage multiorgan diseases. Approved FD-specific treatments include enzyme replacement therapy (ERT) and pharmacological chaperone therapy, while other novel therapeutic approaches, such as substrate reduction therapy, gene therapy, and mRNA-based therapy, are still in development (Table 3) [14,15]. To begin with, ERT is indicated in all symptomatic patients with an established FD diagnosis, which can delay FD advancement and reduce the burden of cardiac events when started at earlier stage [5]. Specifically, in patients with late-onset FD, studies have revealed that ERT decreases LV mass and wall thickness but does not significantly improve heart function (neither systolic nor diastolic), although evidence is limited [5,16]. Most studies further illustrate the poor response of ERT in the heart when applied in the advanced stage, particularly in patients with extensive fibrosis [4,5]. Importantly, cardiac fibrosis stands as a negative cardiac beneficial factor for ERT. Hence, it should be administered as early as possible, before substantial fibrosis development, and the assessment of fibrosis progression by CMR should be considered before and during ERT, in order to evaluate the patient’s prognosis and treatment effectiveness [4,11]. Next, chaperone therapy is only prescribed for patients with amenable *GLA* pathogenic variants, but the effectiveness of this therapy is still debatable. Lastly, other novel treatments, including substrate reduction therapy, result in decreased Gb3 synthesis, which directly lower the cellular load. All gene therapy or mRNA-based therapy aims to restore the defective α-Gal A activity [15,17]. These new therapies will expand in the foreseeable future and hold promise for FD patients. Although FD-specific treatments have changed the natural history of FD, cardiac involvements remain the main prognostic determinant. Updated recommendations on the management of cardiovascular diseases of FD have been published in recent documents [4,5,6].

### 3.4. Management of FD with Symptomatic High-Grade AV Block

Symptomatic high-grade AV block should be treated following the current guidelines [18]. After receiving dual-chamber pacemaker implantation, this patient revealed that he had been in a setting of high-burden RV pacing for years with aggravating heart failure symptoms. A recent study proposed that chronic RV pacing dependence can cause electromechanical dyssynchrony between the ventricles and sequential maladaptive cardiac remodeling. Ultimately, LV systolic dysfunction, as a decrement in LVEF, will be displayed (i.e., pacing-induced cardiomyopathy, PICM) [19]. The most effective treatment for PICM is to upgrade to a physiologic pacing system (e.g., biventricular pacing, His bundle pacing, and left bundle area pacing) [19,20]. After upgradation, a significant improvement of LV systolic function was observed [20].

In FD patients receiving pacemakers, great concern exists about the adverse effect of non-physiological RV pacing with higher risk of developing PICM. From a pathophysiologic perspective, RV pacing contributes to fundamentally altered electrical pattern of ventricular activation. Disturbed electrical activation leads to impaired contraction and redistribution of myocardial strain [19]. These effects cause an abnormal cardiomyocyte metabolism and revised regional perfusion. Besides, in FD patients, deposits (Gb3 and lyzo-Gb3) accumulation and fibrosis replacement within the heart may increase the chance of ventricular dyssynchrony and catalyze the process of cardiac remodeling. For these reasons, physiologic pacing should be initially considered in FD patients who require pacing. However, routine insertion of physiologic pacing systems in patients with low burden of RV pacing (<40%) or preserved systolic function (LVEF > 50%) has not emerged as the standard of care in recent guidelines [18,21]. We strongly suggest that physiologic pacing, rather than traditional RV pacing, be considered for FD patients with symptomatic high-grade AV block, in order to prevent PICM. This novel recommendation still lacks adequate clinical trials to validate the effectiveness in FD patients.

All in all, this article focuses on two key concepts. First, physicians should keep FD in mind when noticing unexplained LVH in combination with specific cardiac alert signs. Earlier detection of the disease and initiation of FD-specific treatments leads to a superior prognosis. Secondly, we have raised a novel and important issue that physiologic pacing systems might play a crucial role in FD patients with AV block requiring pacing, especially in patients who already have impaired LV systolic function. Further, we have provided a comprehensive algorithm (Figure 3), based on elderly patients with unexplained LVH, to guide the diagnostic and therapeutic approaches of FD with symptomatic high-grade AV block, which may be useful to support clinical judgement. Future research should focus on validating the pros and cons of the physiologic pacing system in FD patients and investigate the beneficial effects of FD-specific treatments of the cardiac variants of FD patients specifically.

## 4. Conclusions

Overall, this article focuses on two key concepts. First, physicians should keep FD in mind when noticing unexplained LVH in combination with specific cardiac alert signs. Earlier detection of the disease and initiation of FD-specific treatments leads to a superior prognosis. Second, we have raised a novel issue that physiologic pacing systems play a crucial role in patients with FD who have AV block requiring pacing, especially in patients who already have impaired LV systolic function. Furthermore, we provided a comprehensive algorithm (Figure 2), based on elderly patients with unexplained LVH, to guide the diagnostic and therapeutic approaches of FD with symptomatic high-grade AV block. Future research should focus on validating the pros and cons of the physiologic pacing system in patients with FD and investigating the beneficial effects of FD-specific treatments of the cardiac variants of patients with FD specifically.

## Figures and Tables

**Figure 1 jcm-11-03522-f001:**
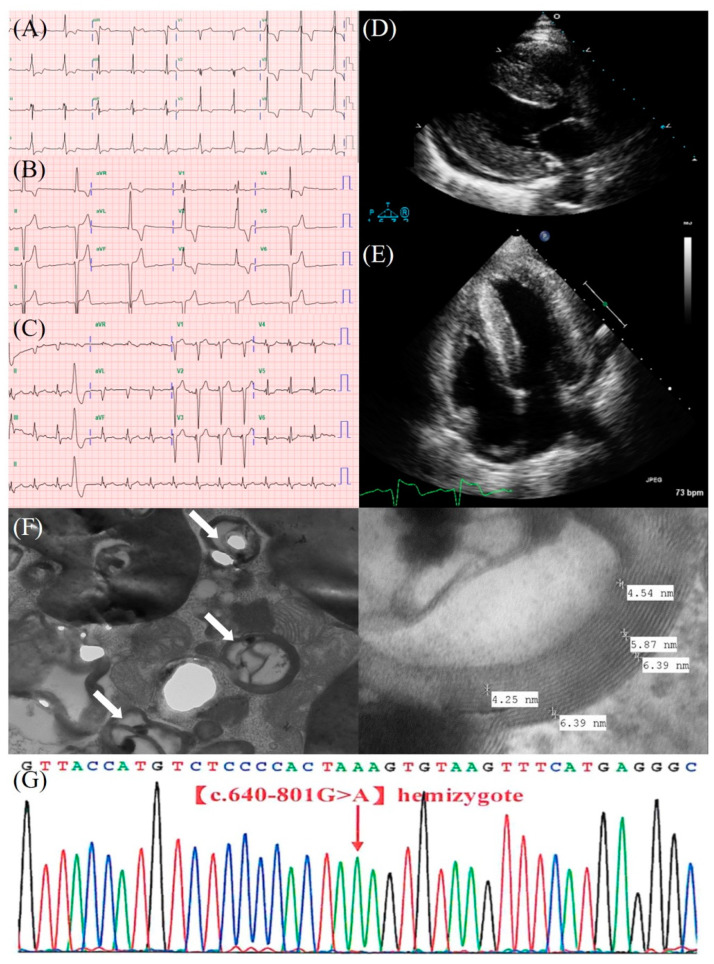
(**A**) The initial 12-lead ECG showed short PR-interval and LVH pattern. (**B**) The 12-lead ECG showed sinus rhythm with 3:1 AV block. (**C**) The 12-lead ECG showed biventricular pacing rhythm. (**D**,**E**) The initial TTE for the patient showed the generalized LVH pattern ((**D**): parasternal long axis view and (**E**): apical four chamber view). (**F**) The pathology showed the typical features of FD on electron microscopy (arrows indicate zebra bodies, with a periodicity of 5–6 nm in the cardiomyocytes). (**G**) The genetic study shows c.640-801G>A polymorphism (cardiac variants of FD).

**Figure 2 jcm-11-03522-f002:**
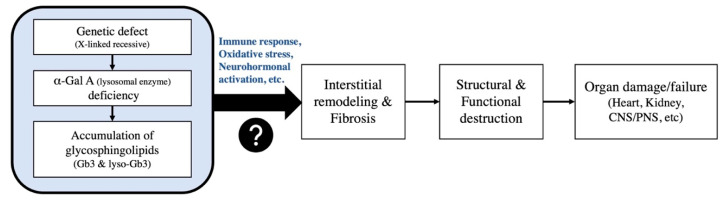
Brief concepts of the pathophysiology of FD [14,15]. α-Gal A: α-galactosidase A, Gb3: globotriaosylceramide, lyso-Gb3: globotriaosylsphingosine, CNS: central nervous system, PNS: peripheral nervous system.

**Figure 3 jcm-11-03522-f003:**
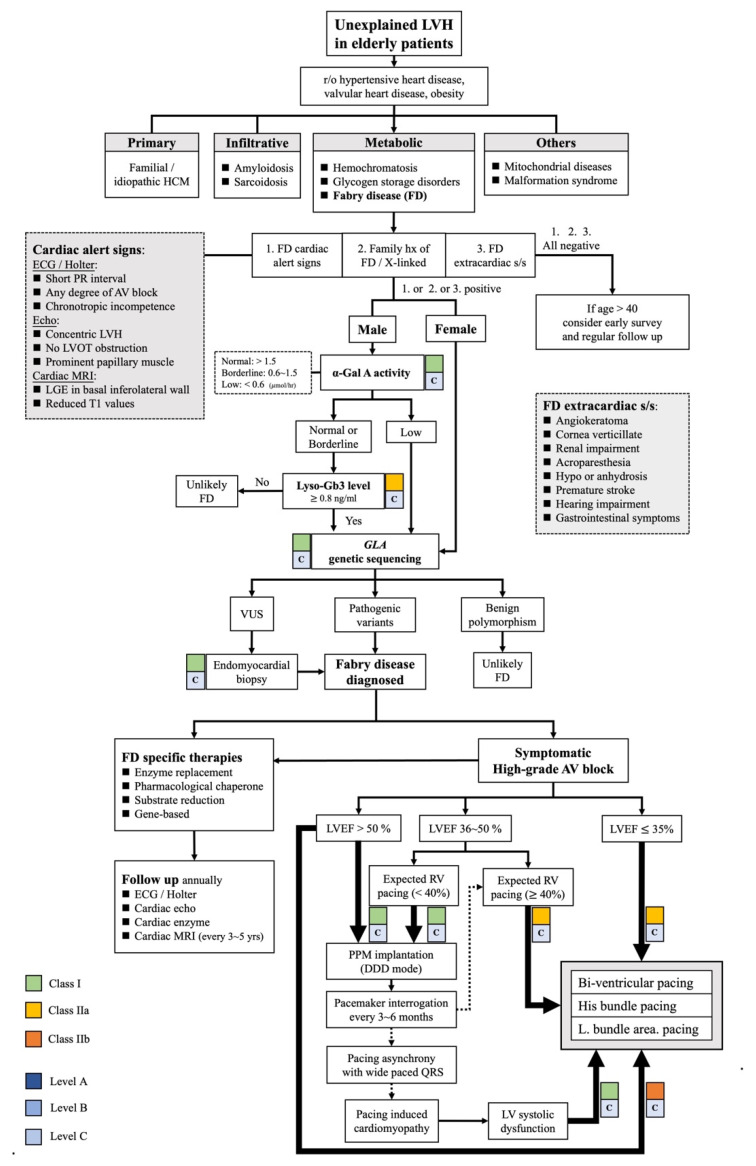
A hypothetical algorithm to evaluate elderly patients who have unexplained LVH to the diagnostic and therapeutic management of FD with high-grade AV block. LVH: left ventricular hypertrophy, r/o: rule out, HCM: hypertrophic cardiomyopathy, FD: Fabry disease, hx: history, LVOT: left ventricular outflow tract, MRI: magnetic resonance imaging, LGE: late gadolinium enhancement, s/s: symptoms/signs, α-Gal A: α-galactosidase A, Lyso-Gb3: globotriaosylsphingosine, *GLA*: α-galactosidase A gene, VUS: variants of unknown significance, RV: right ventricular, LVEF: left ventricular ejection fraction, PPM: permanent pacemaker, DDD: dual-chamber pacing. Classes I, IIa, IIb: the classes of recommendation based on existing studies or guidelines. Levels A, B, C: the levels of evidence based of existing studies [4,18,20,21].

**Table 1 jcm-11-03522-t001:** Clinical history.

Year	Age	Symptoms	Evaluation	Management
2008	63	Palpitation	**ECG**: Sinus rhythm, ventricular preexcitation, LVH (Figure 1A) **Holter**: Normal **TTE**: Concentric LVH, impaired LV relaxation (Figure 1D,E)	OPD follow-up
2011	66	Palpitation and chest tightness	**ECG**: Sinus rhythm, ventricular preexcitation, LVH **Holter**: Normal	OPD follow-up
2015	70	Palpitation and chest tightness	**Holter**: Normal **TTE**: Concentric LVH, LV diastolic dysfunction, adequate LV systolic function, LVEF = 76%, E/e′: 20.2, LV mass index: 273.0 g/m^2^ **Treadmill test**: Positive for ischemia	PCI, OPD follow-up
2016	71	Dizziness	**ECG**: 3:1 AV block (Figure 1B)	PPM (DDDR), OPD follow-up
2017	72	X	**ECG**: Ventricular pacing rhythm, QRS = 180 ms	OPD follow-up
2019	74	DOE	**ECG**: Ventricular pacing rhythm, QRS = 180 ms **TTE**: Concentric LVH, apical LV hypokinesis, borderline LV systolic function, LVEF = 53%, E/e′: 17.0, LV mass index: 292.9 g/m^2^	HF drugs, OPD follow-up
2021	76	Aggravated DOE	**ECG**: Ventricular pacing rhythm, QRS = 200 ms **ECG (s/p CRT)**: Biventricular pacing rhythm, QRS = 160 ms (Figure 1C) **TTE**: Concentric LVH, large apical LV akinesis, impaired LV systolic function, LVEF = 20%, E/e′: 21.4, LV mass index: 160.0 g/m^2^ **TTE (s/p CRT)**: Concentric LVH, apical LV hypokinesis, LVEF = 35%, E/e′: 11.9, LV mass index: 214.5 g/m^2^	CRT, HF drugs, ERT, OPD follow-up

ECG: electrocardiogram, TTE: transthoracic echocardiogram, LVH: left ventricular hypertrophy, LV: left ventricular, OPD: outpatient department, g/m^2^: grams per square meter, LVEF: left ventricular ejection fraction, PCI: percutaneous coronary intervention, HF: heart failure, DOE: dyspnea on exertion, PPM: permanent pacemaker, DDDR: dual-chamber with rate modulation, s/p: status post, CRT: cardiac resynchronization therapy, ERT: enzyme replacement therapy.

**Table 2 jcm-11-03522-t002:** Fabry disease survey.

Examination	Results	Reference Value
α-Gal A activity	1.17 μmol/h (borderline)	N > 1.5 Borderline: 0.6~1.5 (μmol/h)
Plasma Lyso-Gb3	11.95 ng/mL (elevated)	N < 0.8 (ng/mL)
Endomyocardial biopsy	Cardiomyocytes are focally vacuolated with a lace-like appearance. The electron microscope showed laminated lysosomal inclusions (zebra bodies) (Figure 1F).	Compatible with FD
Genetic sequencing	Genotype: c.640-801G>A (Figure 1G)	Also known as IVS4+919G>A and c.936+919G>A, Cardiac variant of FD [7]

α-Gal A: α-galactosidase A, Lyso-Gb3: globotriaosylsphingosine, N: normal value, FD: Fabry disease.

**Table 3 jcm-11-03522-t003:** Diagnosis and management of FD.

**Cardiac Imaging**	**Cardiac Alert Signs [4,13]**
ECG	Short PR interval, AV block, chronotropic incompetence
2D-Echocardiography	Concentric LVH, prominent papillary muscles, diastolic dysfunction
CMR	LGE in the basal inferolateral wall, reduced T1 values
**Diagnostic examinations**	**Diagnostic criteria [3,4,7,12]**
α-Gal A activity	N > 1.5, borderline: 0.6~1.5 (μmol/h)
Plasma Lyso-Gb3	N < 0.8 (ng/mL)
Endomyocardial biopsy	General: diffuse vacuolization with lace-like appearance. The electron microscope: laminated lysosomal inclusions (zebra bodies), focal loss of myofilaments.
Genetic sequencing (Pathogenic variants)	Classical phenotype: more than 840 private mutations, e.g., p.R227X, p.R220X, p.R342X Late-onset phenotype: p.R301Q, p.R363H, p.F113L, p.N215S, IVS4+919G>A
**Disease-specific therapies**	**Detailed information [4,14,15]**
ERT	IV agalsidase alpha and beta (approved), pegunigalsidase alfa (ongoing trials)
Pharmacological chaperone	Oral Miglastat (approved)
SRT	Oral lucerastat, venglustat (ongoing trials)
Gene-based therapy	Gene transfer, mRNA (ongoing trials)

FD: Fabry disease, ECG: electrocardiogram, AV: atrioventricular, 2D: 2 dimensional, LVH: left ventricular hypertrophy, CMR: cardiac magnetic resonance imaging, LGE: late gadolinium enhancement, α-Gal A: α-galactosidase A, Lyso-Gb3: globotriaosylsphingosine, N: normal value, ERT: enzyme replacement therapy, IV: intravenous administration, SRT: substrate reduction therapy, mRNA: messenger ribonucleic acid.

## Data Availability

The datasets used during the current study are available from the corresponding author on reasonable request.

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
