# Peer review of "Unexplained Left Ventricular Hypertrophy with Symptomatic High-Grade Atrioventricular Block in Elderly Patients: A Case Report"

_jcm, 2022, doi:10.3390/jcm11123522_

Round 1

Reviewer 1 Report

This report describes an interesting case of Fabry`s disease with left ventricular hypertrophy and atrioventricular block in an older patient 76 years of age.

The descripition of the case is comprehensive as is the presentation of this severe genetic disease. There are only some minor comments to improve readability of the text and attraction of potentially interested readers.

1. Abstract line 12: check ".....symptomatic high-grade atrioventricular (AV)"....?? 

2. Abstract lines 15 - 17. Check "English wording" of the sentence:  "Further FD surveys included performing biochemical examinations....."

3. Introduction, line 30: please provide some citations with respect to the causes of LVH

4. Introduction: Provide some data of the prevalence and incidence of Fabry disease

5. Introduction, line 37: what ist meant by an "effective treatment", please gives some more informations.

6. Case presentation, line 68: what is meant by "presiding rhythm"`? you may replace this term by "predominant rhythm"

7. please explain the abbreviation "OPD" not only within the figure legend, but also within the official list of abbreviations line 292

8. Discussion, line 110: it is unclear what is meant by "... with limited cardiac diseases...", check English language

9. Discussion, lines 144 and above: In general it is difficult for the reader to follow an extended text passage describing complicated and detailled pathphysiological pathways. I therefore strongly recommend to shorten the text but provide a comprehensable flow-chart or figure to present and explain the pathophysiology of FD. 

10. Discussion, 3.2/3.3. Diagnosis of FD: provide a table for presenting the diagnostic features. This also applies for the presentation and description of the "FD managament".

11. Discussion, 3.4, line 237: the English language should be improved.

Author Response

Dear Editor & Reviewers

Thank you for reviewing our manuscript titled: “Unexplained Left Ventricular Hypertrophy with Symptomatic High-grade Atrioventricular Block in Elderly patients: a Case report”. We greatly appreciate your valuable comments and conditionally accept this manuscript. We have carefully reviewed and revised the manuscript and feel that it has strengthened the article. Also, we appreciate that this article aroused your interest in raising valuable comments. Please find our point-by-point responses to the Reviewer’s comments listed below.

Thank you for all your comments and the opportunity to have our article published in the Journal of Clinical Medicine. We hope that our careful revisions meet your recommendations.

With our sincere thanks,

Responses to Reviewer 1

Point 1. Abstract, line 12: check ".....symptomatic high-grade atrioventricular (AV)"....??

Response 1. Abstract, line 12: The sentence was revised by ".....symptomatic high-grade atrioventricular (AV) block"

Point 2. Abstract, lines 15 - 17. Check "English wording" of the sentence:  "Further FD surveys included performing biochemical examinations....."

Response 2. Abstract, lines 16-17: The sentence was revised by " Further FD surveys were performed included biochemical examinations....."

Point 3. Introduction, line 30: please provide some citations with respect to the causes of LVH

Response 3. Introduction, line 30: We’ve added 2 citations for the possible causes of LVH (DOI: 10.1016/j.jchf.2018.02.010, DOI: 10.1056/NEJMoa033349) and inserted totally 3 citations (doi:10.1093/eurheartj/ehs166).

Point 4. Introduction: Provide some data of the prevalence and incidence of Fabry disease

Response 4. Introduction, line 36: We’ve provided the prevalence and incidence of Fabry disease.

Point 5. Introduction, line 37: what ist meant by an "effective treatment", please gives some more information.

Response 5. Introduction, line 38: The detailed information of the effective treatment was given in the Discussion 3.3 Management of FD, such as disease specific treatment (enzyme replacement therapy and pharmacological chaperone therapy) and certain therapies to manage multiorgan diseases.

Point 6. Case presentation, line 68: what is meant by "presiding rhythm"`? you may replace this term by "predominant rhythm"

Response 6. Case presentation, line 69: We’ve revised this term by "predominant rhythm".

Point 7. please explain the abbreviation "OPD" not only within the figure legend, but also within the official list of abbreviations line 292

Response 7. The official list of abbreviations, line 306: The abbreviation, OPD, which stands for outpatient department was noted on the list.

Point 8. Discussion, line 110: it is unclear what is meant by "... with limited cardiac diseases...", check English language

Response 8. Discussion, lines 111-112: The sentence was revised by " In the present case, the patient presented with late onset LVH with symptomatic high-grade AV block and without extracardiac comorbidities…".

Point 9. Discussion, lines 144 and above: In general it is difficult for the reader to follow an extended text passage describing complicated and detailled pathphysiological pathways. I therefore strongly recommend to shorten the text but provide a comprehensable flow-chart or figure to present and explain the pathophysiology of FD.

Response 9. The detailed pathophysiology pathways of FD were still under investigated, thus we’ve condensed the sentences outlining the core features of FD in Discussion 3.1, lines 146-150 and provided a brief concept of the pathophysiology of FD in Figure 2 (line 180).

Point 10. Discussion, 3.2/3.3. Diagnosis of FD: provide a table for presenting the diagnostic features. This also applies for the presentation and description of the "FD management".

Response 10. We’ve provided Table 3 (line 213) for presenting the diagnosis and management of FD.

Point 11. Discussion, 3.4, line 237: the English language should be improved.

Response 11. Discussion, 3.4, line 246: The sentence was revised by "Symptomatic high-grade AV block should be treated following the current guidelines".

Reviewer 2 Report

This manuscript provides a clinical example of the diagnosis of Fabry disease in a 76-year-old patient with unclear left ventricular hypertrophy and the development of complete atrioventricular block. In addition, the authors proposed a diagnostic and therapeutic algorithm based on this clinical case and analysis of literature data.

This manuscript may be of some interest to clinicians, but I had a number of comments while reviewing it.

1.      The concept of the article is not fully understood. For a clinical case, it contains little clinical data about the patient (for example, there are no detailed protocols of the results of transthoracic echocardiography over the years of observation: there are no data on the thickness of the left ventricle walls, on the left ventricle myocardium mass, on the indicators of left ventricle diastolic function, etc.). For a review of the literature, the number of references is clearly small.

2.      It seems to me that undoubtedly deserves discussion in this clinical case - why the diagnosis of Fabry disease was made only after 14 years of observation, although specific signs of heart damage (shortened PR interval, unexplained left ventricular hypertrophy) were determined in the patient already in 2008. Therefore, it would be interesting to analyze the literature data on similar clinical cases of late manifestation of Fabry disease, and possible diagnostic algorithms for its early diagnosis (for example - doi: 10.1016/j.jcmg.2017.10.018). Moreover, despite the rare occurrence of Fabry disease, a large number of detailed reviews on heart damage in this pathology have been presented in the recent scientific literature (in addition to the review authors cited - doi: 10.1016/j.jacc.2020.12.024 and doi: 10.1016/j.acvd.2019.01.002, other reviews can be mentioned - doi: 10.3390/cells10061532, doi: 10.3390/jcm10143026, and doi: 10.3390/jcm10091994).

3.      The hypothetical algorithm proposed by the authors also raises a number of questions. Why are Fabry disease and cardiac signs of anxiety classified as IIb at the top of Figure 2 (judging by the font color)? The suggestion of a treatment strategy at the bottom of the figure, based on only a single clinical case, does not seem to be entirely justified. In addition, if the authors of the algorithm suggest using the recommendation classes (I, IIa, and IIb), then the level of evidence (A, B, and C) should also be indicated.

References:

Jain R, Kalvin L, Johnson B, Muthukumar L, Khandheria BK, Tajik AJ. Many Faces of Fabry's Cardiomyopathy. JACC Cardiovasc Imaging. 2018 Apr;11(4):644-647. doi: 10.1016/j.jcmg.2017.10.018.

Pieroni M, Moon JC, Arbustini E, Barriales-Villa R, Camporeale A, Vujkovac AC, Elliott PM, Hagege A, Kuusisto J, Linhart A, Nordbeck P, Olivotto I, Pietilä-Effati P, Namdar M. Cardiac Involvement in Fabry Disease: JACC Review Topic of the Week. J Am Coll Cardiol. 2021 Feb 23;77(7):922-936. doi: 10.1016/j.jacc.2020.12.024.

Yim J, Yau O, Yeung DF, Tsang TSM. Fabry Cardiomyopathy: Current Practice and Future Directions. Cells. 2021 Jun 17;10(6):1532. doi: 10.3390/cells10061532.

Vardarli I, Weber M, Rischpler C, Führer D, Herrmann K, Weidemann F. Fabry Cardiomyopathy: Current Treatment and Future Options. J Clin Med. 2021 Jul 7;10(14):3026. doi: 10.3390/jcm10143026.

Esposito R, Santoro C, Mandoli GE, Cuomo V, Sorrentino R, La Mura L, Pastore MC, Bandera F, D'Ascenzi F, Malagoli A, Benfari G, D'Andrea A, Cameli M. Cardiac Imaging in Anderson-Fabry Disease: Past, Present and Future. J Clin Med. 2021 May 6;10(9):1994. doi: 10.3390/jcm10091994.

Hagège A, Réant P, Habib G, Damy T, Barone-Rochette G, Soulat G, Donal E, Germain DP. Fabry disease in cardiology practice: Literature review and expert point of view. Arch Cardiovasc Dis. 2019 Apr;112(4):278-287. doi: 10.1016/j.acvd.2019.01.002.

Author Response

Dear Editor & Reviewers

Thank you for reviewing our manuscript titled: “Unexplained Left Ventricular Hypertrophy with Symptomatic High-grade Atrioventricular Block in Elderly patients: a Case report”. We greatly appreciate your valuable comments and conditionally accept this manuscript. We have carefully reviewed and revised the manuscript and feel that it has strengthened the article. Also, we appreciate that this article aroused your interest in raising valuable comments. Please find our point-by-point responses to the Reviewer’s comments listed below.

Thank you for all your comments and the opportunity to have our article published in the Journal of Clinical Medicine. We hope that our careful revisions meet your recommendations.

With our sincere thanks,

Responses to Reviewer 2

Point 1.The concept of the article is not fully understood. For a clinical case, it contains little clinical data about the patient (for example, there are no detailed protocols of the results of transthoracic echocardiography over the years of observation: there are no data on the thickness of the left ventricle walls, on the left ventricle myocardium mass, on the indicators of left ventricle diastolic function, etc.). For a review of the literature, the number of references is clearly small.

Response 1. Sincerely thanks for the recommendation. In light of the numerous reviews on cardiac imaging in FD which have been presented recently. We’d like to highlight the other issues from electrophysiological perspectives. The main concept of this article was to raise an idea that physiologic pacing systems play a crucial role in patients with FD who have AV block requiring pacing, especially in patients who already have impaired LV systolic function. Therefore, the detailed echocardiographic findings were not being scrutinized initially, and the standardized protocol of the echocardiography in our hospital wasn’t well established before 2015. We reviewed the medical charts again and chose the specific parameters for evaluating FD cardiomyopathy on echocardiography, the left ventricle myocardium mass index and E/e’ ratio (doi:10.3390/jcm10091994), which were added in Table 1. Unfortunately, these parameters weren’t recorded in the report in 2008, thus, the initial information of the patient was lacking.

The title was revised to remove the setting of the literature review.

Point 2. It seems to me that undoubtedly deserves discussion in this clinical case - why the diagnosis of Fabry disease was made only after 14 years of observation, although specific signs of heart damage (shortened PR interval, unexplained left ventricular hypertrophy) were determined in the patient already in 2008. Therefore, it would be interesting to analyze the literature data on similar clinical cases of late manifestation of Fabry disease, and possible diagnostic algorithms for its early diagnosis (for example - doi: 10.1016/j.jcmg.2017.10.018). Moreover, despite the rare occurrence of Fabry disease, a large number of detailed reviews on heart damage in this pathology have been presented in the recent scientific literature (in addition to the review authors cited - doi: 10.1016/j.jacc.2020.12.024 and doi: 10.1016/j.acvd.2019.01.002, other reviews can be mentioned - doi: 10.3390/cells10061532, doi: 10.3390/jcm10143026, and doi: 10.3390/jcm10091994).

Response 2. Thanks for the comments which were appeared to be highly essential for us. Originally, this patient was regularly followed up in Family physicians due to recurrent palpitations from 2008 to 2015. Although, the image interpretations had already revealed some cardiac alert signs in 2008, no prominent symptoms were noted to raise further surveys. In 2016, the patient began to display obscure cardiac diseases and was referred to our Cardiology department for further evaluation. Since then, he was closely followed up in our department. In 2019, he presented symptomatic high-grade AV block following rapid decline of left systolic function after pacemaker implantation. Extra investigations were subsequently launched based on the available examinations in our hospital at that time, and he was ultimately diagnosed with cardiac variant of FD in 2021.

The reviews which you recommended were inserted in the article to offer more comprehensive views, sincerely thanks!

Point 3. The hypothetical algorithm proposed by the authors also raises a number of questions. Why are Fabry disease and cardiac signs of anxiety classified as IIb at the top of Figure 2 (judging by the font color)? The suggestion of a treatment strategy at the bottom of the figure, based on only a single clinical case, does not seem to be entirely justified. In addition, if the authors of the algorithm suggest using the recommendation classes (I, IIa, and IIb), then the level of evidence (A, B, and C) should also be indicated.

Response 3. The font color of “Fabry disease and cardiac alert signs” were slightly different from the IIb recommendation classes but was easily confusing. We’ve changed the font color to prevent confusion. Additionally, we adjusted the algorithm to provide clearer version and added the level of evidence. (Figure 3, line 283)

Round 2

Reviewer 2 Report

I am satisfied with the work done by the authors on the manuscript and the responses to my comments. I have no other comments.